# Macrophages Promote Subtype Conversion and Endocrine Resistance in Breast Cancer

**DOI:** 10.3390/cancers16030678

**Published:** 2024-02-05

**Authors:** Xiaoyan Zhang, Fengyu Yang, Zhijian Huang, Xiaojun Liu, Gan Xia, Jieye Huang, Yang Yang, Junchen Li, Jin Huang, Yuxin Liu, Ti Zhou, Weiwei Qi, Guoquan Gao, Xia Yang

**Affiliations:** 1Department of Biochemistry, Zhongshan School of Medicine, Sun Yat-sen University, Guangzhou 510080, China; zhangxy529@mail2.sysu.edu.cn (X.Z.); yangfy9@mail2.sysu.edu.cn (F.Y.); huangzhijian@sysush.com (Z.H.); liuxj87@mail2.sysu.edu.cn (X.L.); xiag@mail2.sysu.edu.cn (G.X.); huangjy86@mail2.sysu.edu.cn (J.H.); yangy639@mail2.sysu.edu.cn (Y.Y.); lijch75@mail2.sysu.edu.cn (J.L.); huangj553@mail2.sysu.edu.cn (J.H.); liuyx339@mail2.sysu.edu.cn (Y.L.); zhouti2@mail.sysu.edu.cn (T.Z.); qiww3@mail.sysu.edu.cn (W.Q.); 2Department of Pathology, The Seventh Affiliated Hospital of Sun Yat-sen University, Shenzhen 518107, China; 3Department of Internal Medicine, Guangzhou Women and Children’s Medical Center, Guangzhou Medical University, Guangzhou 510700, China; 4Guangdong Province Key Laboratory of Brain Function and Disease, Zhongshan School of Medicine, Sun Yat-sen University, Guangzhou 510080, China; 5Guangdong Engineering & Technology Research Center for Gene Manipulation and Biomacromolecular Products, Sun Yat-sen University, Guangzhou 510080, China

**Keywords:** breast cancer, clinical subtype, endocrine resistance, macrophage, MNX1

## Abstract

**Simple Summary:**

Breast cancer is a highly heterogeneous disease on the molecular level. The molecular subtype will be altered during metastasis and affect patient outcomes and treatments. Breast cancer is prone to lymph node metastasis. Our study aimed to investigate whether subtype conversion occurs during lymph node metastasis and its underlying mechanism. We confirmed that hormone receptors were down-regulated, while HER2 was up-regulated during lymph node metastasis, and macrophages played a significant role in the process, probably via MNX1. Targeting macrophages or MNX1 may provide new avenues for endocrine therapy and targeted treatment of breast cancer patients with lymph node metastasis.

**Abstract:**

Background: The progression of tumors from less aggressive subtypes to more aggressive states during metastasis poses challenges for treatment strategies. Previous studies have revealed the molecular subtype conversion between primary and metastatic tumors in breast cancer (BC). However, the subtype conversion during lymph node metastasis (LNM) and the underlying mechanism remains unclear. Methods: We compared clinical subtypes in paired primary tumors and positive lymph nodes (PLNs) in BC patients and further validated them in the mouse model. Bioinformatics analysis and macrophage-conditioned medium treatment were performed to investigate the role of macrophages in subtype conversion. Results: During LNM, hormone receptors (HRs) were down-regulated, while HER2 was up-regulated, leading to the transformation of luminal A tumors towards luminal B tumors and from luminal B subtype towards HER2-enriched (HER2-E) subtype. The mouse model demonstrated the elevated levels of HER2 in PLN while retaining luminal characteristics. Among the various cells in the tumor microenvironment (TME), macrophages were the most clinically relevant in terms of prognosis. The treatment of a macrophage-conditioned medium further confirmed the downregulation of HR expression and upregulation of HER2 expression, inducing tamoxifen resistance. Through bioinformatics analysis, MNX1 was identified as a potential transcription factor governing the expression of HR and HER2. Conclusion: Our study revealed the HER2-E subtype conversion during LNM in BC. Macrophages were the crucial cell type in TME, inducing the downregulation of HR and upregulation of HER2, probably via MNX1. Targeting macrophages or MNX1 may provide new avenues for endocrine therapy and targeted treatment of BC patients with LNM.

## 1. Introduction

Until now, breast cancer has been the most common type of malignant tumor in women worldwide [1,2]. Approximately 70–80% of patients with early-stage, non-metastatic breast cancer can be cured, while metastatic breast cancer is considered an incurable disease and the leading cause of cancer-related death [3,4]. Breast cancer is a highly heterogeneous disease on the molecular level [5,6,7,8]. Based on the immunohistochemistry (IHC) expression of estrogen receptor (ER), progesterone receptor (PR), and human epidermal growth factor receptor 2 (HER2), BC is classified into hormone receptor-positive BC (luminal), constituting approximately 70% of BC instances; HER2-enriched BC (HER2-E), accounting for 15% to 20% of BC cases; and triple-negative BC (TNBC), which makes up 15% of BC cases [9,10,11,12]. Breast cancer subtypes are associated with significant differences in clinical outcomes. Patients with luminal tumors had a favorable prognosis, while patients with HER2-E and TNBC breast cancer experienced worse outcomes [13,14].

Previous studies have revealed changes in clinical biomarkers (ER, PR, and HER2) between primary and metastatic tumors [4,15,16,17,18,19]. Based on some retrospective studies, the discordance rates of ER, PR, and HER2 exhibited considerable variation among breast cancer patients, ranging from 3% to 54% for ER, 5% to 78% for PR, and 0% to 34% for HER2 [19,20,21,22]. This phenomenon is termed “receptor conversion” [16]. While molecular subtypes of breast cancer are largely maintained during metastatic progression, luminal A breast cancer is an exception, which can convert to luminal B or HER2-E breast cancer [4,23]. Outcomes and treatment options for breast cancer differ according to molecular subtypes [9,24]. Resistance to HR- or HER2-targeted therapies or the absence of these receptors are major therapeutic concerns [25].

As an essential component of TME, tumor-associated macrophages (TAMs) can render patients resistant to chemotherapy agents and checkpoint-blocking immunotherapy [26,27,28]. TAMs play a critical role in endocrine therapy resistance in breast cancer patients. Macrophages can cause the loss of ERα expression by the direct chromatin action of the c-Jun/ERK2 complex [29] or the inactivation of FOXO3a [30]. TAMs also can induce endocrine therapy resistance by hyperphosphorylation of ERα through the NF-κB/STAT3/ERK pathway [31]. Macrophage-induced expression of EGFR may lead to tamoxifen resistance in breast cancer [32]. The PI3K/Akt/mTOR signaling pathway is a dominant factor in the endocrine resistance of breast cancer [33,34,35]. TAMs can promote tamoxifen resistance in BC by activating the PI3K/Akt/mTOR pathway [36,37]. In contrast, the relationship between TAMs and HER2 expression remains unknown. Lymph node metastasis is the early metastatic mode of breast cancer, especially for luminal breast cancer [3]. Several extensive cohort studies have indicated that lymph node metastases are associated with poor clinical outcomes [38,39,40]. Previous studies mainly focused on changes in distant organ metastases, but the early changes in positive lymph nodes also need more attention.

Endocrine therapy resistance poses a major clinical challenge in breast cancer patients. Further research is still needed to investigate the underlying mechanism between TAMs and endocrine therapy resistance. Additionally, the role TAMs play in “receptor conversion” requires exploration. A wealth of evidence highlights significant differences in tissue characteristics between primary breast tumors and paired metastases. Therapies effective for primary tumors may not be suitable for metastases. Furthermore, we are committed to exploring the role of macrophages in subtype conversion between primary breast tumors and lymph node metastases.

## 2. Material and Methods

### 2.1. Human Samples

Sixty-four human BC samples with complete medical records were collected from Sun Yat-sen University Cancer Center from 1 January 2009 to 31 December 2015. Informed consent from the patients was obtained before surgery, and the Ethics Committee of Sun Yat-sen University Cancer Center approved the use of medical records and histological slides. The BC tissue microarrays were purchased from Guangzhou Wozhao Biotech Co., Ltd., Guangzhou, China (BR10010e1, 50 cases) with detailed clinical information. All procedures were carried out under a consensus agreement and complied with the requirements of the Chinese Ethics Review Committee. The research methodology complied with the criteria set out in the Declaration of Helsinki. The clinical and biological characteristics of the patients are shown in Appendix A.

### 2.2. Cell Lines and Culture

The human BC cell lines (T47D, BT474) were provided by Professor Erwei Song from Sun Yat-sen Memorial Hospital, Sun Yat-sen University. T47D was maintained in DMEM (12800082, Gibco, Waltham, MA, USA) containing 10% FBS (FSP500, ExCell Bio, Shanghai, China), and BT474 was maintained in RIPM-1640 (C22400500CP, Gibco, Waltham, MA, USA) supplemented with 10% FBS. All BC cell lines were incubated at 37 °C under a humidified atmosphere with 5% CO_2_.

### 2.3. Western Blotting

Total proteins were collected by SDS lysis buffer supplemented with 1 × protease inhibitor (HY-K0010, MedChem Express, Middlesex, NJ, USA), and protein concentrations were quantified by the BCA assay kit (KGP902, KeyGen, Nanjing, China). 10% SDS-PAGE separated the proteins, then transferred them to 0.45 μm polyvinyl difluoride (PVDF) membranes (IPVH00010, Millipore, Boston, MA, USA). The PVDF membranes were blocked in 7% skimmed milk supplemented with 1× TBST and then incubated overnight at 4 °C with primary antibodies. The following primary antibodies were used: estrogen receptor alpha (sc-8002, Santa Cruz Biotechnology, Dallas, TX, USA, 1:1000), progesterone receptor (DH0001, Abnova, Taipei, Taiwan, 1:2000), HER2 (4290S, Cell Signaling Tech, Boston, MA, USA, 1:2000), and GAPDH (60004-1-Ig, Proteintech, Rosemont, IL, USA, 1:5000). HRP-conjugated anti-rabbit IgG (SA00001-2, Proteintech, Rosemont, IL, USA, 1:2000) and anti-mouse IgG (SA00001-1, Proteintech, Rosemont, IL, USA, 1:2000 for ERα and PR, 1:5000 for GAPDH) were secondary antibodies. Proteins were determined using ECL Plus Reagent(WBKLS0100, Millipore, Boston, MA, USA).

### 2.4. RNA Isolation and RT-qPCR

The total RNA of the BC cells was extracted according to the manufacturer’s instructions for the TRIzol reagent. Gene expression validation by RT-qPCR was performed as previously described [41]. The PCR primer sequences are listed in Appendix A.

### 2.5. Immunohistochemistry Staining

Immunohistochemistry was performed per a standard protocol described previously [41]. The slides were incubated with primary antibodies at 4 °C overnight. The following primary antibodies were used: estrogen receptor alpha (sc-8002, Santa Cruz Biotechnology, Dallas, TX, USA, 1:200), progesterone receptor (DH0001, Abnova, Taipei, Taiwan, 1:200), HER2 (4290S, Cell Signaling Tech, Boston, MA, USA, 1:200) and CD68 (ab201340, Abcam, Cambridge, UK, 1:200). On the second day, the slides were treated with HRP-conjugated secondary antibody. The antigen–antibody complex was visualized by incubation with the DAB IHC detection kit. The slides were photographed through a slide scanner (Axio Scan. Z1, ZEISS, Oberkochen, Germany). Histological ER, PR, and HER2 diagnoses were assessed by 2 pathologists. Clinical subtype conversion refers to the changes in IHC expression levels of ER, PR, or HER2 between breast cancer primary diseases and paired lymph node metastases. Samples with 1% to 100% of tumor nuclei positive for ER or PR were defined as positive. Samples were considered negative for ER or PR if <1% or 0% of tumor cell nuclei were stained. The degree of ER and PR immunostaining was determined by the staining index (SI). The SI was calculated by multiplying the grade of tumor cell proportions by the staining intensity score. The grade of tumor cell proportions was defined as follows: 0, <1% tumor nuclei positive; 1, 1–10% tumor nuclei positive; 2, 10–50% tumor nuclei positive; 3, 51–75% tumor nuclei positive; and 4, >75% tumor nuclei positive. The staining intensity score was defined as follows: 0, no tumor nuclei staining (negatively stained); 1, weak tumor nuclei staining (light yellow); 2, moderate tumor nuclei staining (yellow-brown); and 3, strong tumor nuclei staining (brown). The degree of HER2 immunostaining was graded as follows: 0: no staining or ≤10% of infiltrating cancer cells showed incomplete and weak cell membrane staining; 1+: > 10% of infiltrating cancer cells exhibited incomplete and weak cell membrane staining; 2+: > 10% of infiltrating cancer cells showed weak to moderate strength and intact cell membrane staining or ≤10% of infiltrating cancer cells showed strong and intact cell membrane staining; and 3+: >10% of infiltrating cancer cells showed strong, complete and uniform cell membrane staining. Samples were classified as ‘HER2-negative’ (IHC 0, 1+ or 2+/ISH not-amplified) and ‘HER2-positive’ (IHC 2+/ISH-amplified or 3+). FISH was detected by Sun Yat-sen University Cancer Center. Immunohistochemical staining of CD68 was according to a protocol as described previously [42], and was quantitatively categorized as a score of 0, 1+, 2+, or 3+ for no CD68^+^ cells, ≤10 CD68^+^ cells, 11-20 CD68^+^ cells or ≥21 CD68^+^ cells in the observing field, respectively, at ×200 magnification.

### 2.6. Mice and Tumor Models

All animal studies were reviewed and approved by the Institutional Animal Care and Use Committee of Sun Yat-sen University. Five-week-old specific pathogen-free female BALB/c-nu/nu mice were purchased from Beijing Vital River Laboratory Animal Technology Co., Ltd.,Beijing, China. The footpad implantation model was conducted according to the previous reference [43]. T47D cells were previously modified with luciferase genes (T47D-Luc). For the T47D-Luc cells footpad implantation model, 2 × 10^6^ T47D-Luc cells in 50 μL of PBS were subcutaneously implanted into the footpad area of the hind limb of mice. Twenty-eight days after implantation, the footpad and popliteal LN were imaged with the IVIS Spectrum Imaging System (IVIS Spectrum, Perkin Elmer, Waltham, MA, USA) after intraperitoneal injection of D-luciferin (122799, Perkin Elmer, Waltham, MA, USA, 150 mg/kg mice weight, dissolved with PBS). The primary tumors in the footpad and popliteal LN were dissected for further analyses and experiments.

### 2.7. Macrophage Polarization

THP-1 cells were centrifugally suspended and inoculated on a 6-well plate with 8 × 10^5^ cells per well. The cells were treated with 200 ng/mL PMA (P8139-1MG, Sigma-Aldric, St. Louis, MO, USA, dissolved with DMSO) for 6 h to induce differentiation into macrophages. Then add 100 ng/mL LPS (L2880-10 mg, Sigma-Aldric, St. Louis, MO, USA, dissolved with PBS), polarized it towards M1 or 20 ng/mL IL-4 (200-04-5, PeproTech, Waltham, MA, USA, dissolved with PBS) polarized it towards M2, and continued incubation for 48 h. After the polarization was completed, the cells were washed three times with 1 × PBS to clean the substances that stimulated differentiation, and then 2 mL of RIPM-1640 containing 10% serum was added for re-suspension. The supernatant’s culture was collected for further experiments, and the impurities, such as cell debris, were removed with a filter with a 0.22 μM aperture (SLGP033RB, Millipore, Boston, MA, USA).

### 2.8. Cell Viability Assay

The viability of BC cells was measured through the Cell Counting Kit-8 (CCK-8) assay (CK04, Dojindo, Kumamoto, Japan), following the manufacturer’s protocol. 8000 cells in 100 μL of medium per well were seeded in 96-well plates (701001, NEST, Wuxi, China). According to the experimental requirements, 10 μL of CCK-8 was added to each well after 24, 48, and 72 h. OD values were determined by absorbance at 450 nm using the Sunrise microplate reader (TECAN, Maannedorf, Switzerland) after 2 h incubation in a humidified incubator containing 5% CO_2_ at 37 °C.

### 2.9. Apoptosis Assay

Apoptosis was detected by Annexin V, FITC Apoptosis Detection Kit (DOJINDO, AD10) according to the manufacturer’s protocol. 5 × 10^5^ BC cells in 2 mL of medium per well were seeded in 6-well plates (703001, NEST, Wuxi, China). Treatments of BC cells were added as indicated in the figure legends. 4-Hydroxytamoxifen (S7827, Selleck, Houston, TX, USA, dissolved with DMSO) and Fulvestrant (S1191, Selleck, Houston, TX, USA, dissolved with DMSO) were purchased from Selleck. The cells were washed with PBS and then digested with EDTA-free trypsin solution. The cells were washed twice with PBS after centrifugation. Cells were resuspended with a final concentration of 1 × 10^6^ cells/mL in 1 × binding buffer, 100 μL cell suspension was added with 5 μL Annexin V and 5 μL PI Solution, then cultured for 15 min away from light at room temperature. The apoptosis level was measured by flow cytometry (CytoFLEX, Beckman Coulter, Brea, CA, USA).

### 2.10. Statistical Analysis

The mean and standard deviation (mean ± SD) were used to present all the data. The Student’s *t*-test was used to evaluate the data using GraphPad Prism 8.3.0 software, and a p-value of less than 0.05 determined statistical significance. The Sankey diagram was also drawn by Sangerbox 3.0 [44]. Multivariate Cox hazards regression analysis adjusted for predetermined factors, including age, gender, molecular subtype, treatment, and TNM stage. TIMER 2.0 [45] and EPIC were used to calculate the composition and enrichment of immune cells in TME. The survival analysis between the infiltration degree of macrophages and clinical outcomes was proceeded by TIMER 2.0, R 4.3.1 software, and KM-Plotter. TIMER 2.0 performed the Cox Proportional Hazard Model, and covariates were adjusted, including age, gender, stage, race, and tumor purity. R 4.3.1 software and KM-Plotter performed Kaplan–Meier survival curves and Log-rank statistical analysis. The R packages “survival” and “survminer” were used to analyze the relationship between the degree of macrophage infiltration or gene expression and clinical prognosis. Statistical analyses were conducted with R 4.3.1 and RStudio (2023.06.1 Build 524) software, with packages available through Bioconductor. The correlation matrix displayed the correlation coefficient between MNX1 or NKX2-2 and ESR1, ESR2, ERBB2, and EGFR by employing the packages “ggplot2” and “ggstatsplot”. Statistical significance was determined by *p* < 0.05.

### 2.11. Data Availability

The RNA sequence data and clinical information of breast cancer patients were searched out from the TCGA database (https://portal.gdc.cancer.gov/ (accessed on 17 July 2023)) and the METABRIC database (https://www.cbioportal.org/study/summary?id=brca_metabric (accessed on 17 July 2023)). The PAM50 features of breast cancer patients were acquired from the GEO database (GSE92977).

## 3. Results

### 3.1. Subtype Concordance between Primary Tumors and PLN

We conducted a retrospective analysis of clinical data from two patient cohorts comprising 47 and 64 individuals to investigate the concordance of molecular subtypes between primary tumors and PLN. The baseline characteristics of patients were described in Appendix A. Analyzing the clinical information from the tissue microarrays of 47 breast cancer patients, we observed high subtype concordance for TNBC (92.9%). Among luminal A primary tumors, 6.7% switched to luminal B, and 6.7% converted to TNBC. Similarly, for HER2-E primary tumors, 18.2% switched to TNBC (Figure 1A). Moreover, clinical data of 64 breast cancer patients we collected from the hospital indicated that 17.2% of luminal A primary tumors switched to luminal B, and 13.8% converted to TNBC. 7.7% of luminal B primary tumors converted to HER2-E and 7.7% to TNBC. As for HER2-E primary tumors, 16.7% transformed into TNBC. Remarkably, more aggressive breast cancer subtypes in PLN could revert to subtypes with better prognoses. In PLN, 23.1% of luminal B primary tumors converted to luminal A. HER2-E primary tumors switched to luminal A in 8.3% of cases and to luminal B in 41.7%. 50% of TNBC primary tumors converted to luminal A and 10% to luminal B (Figure 1B). Furthermore, we also made a comparison of PAM50 distribution between primary tumors and positive lymph nodes in the GEO dataset (GSE92977). The distribution of the PAM50 intrinsic subtype in primary tumor versus PLN was 25% versus 0% for normal, 25% versus 8% for luminal A, 21% versus 46% for luminal B, 17% versus 29% for HER2-E and 12% versus 17% for basal-like tumors (Figure 1C). To further validate these findings, we established a popliteal lymph node metastasis model. Luminal A breast cancer cells (T47D) were inoculated on the footpad (Figure 1D). The results indicated upregulation of HER2 protein in four out of eight positive popliteal lymph nodes. Conversely, ER and PR proteins demonstrated downregulation in PLN, with at least one remaining positive (Figure 1E,F).

### 3.2. Macrophages Are Associated with Poor Prognosis and May Contribute to Subtype Conversation

The TME encompasses a variety of immune cell types, such as T cells, B cells, pericytes, tumor-associated macrophages, cancer-associated fibroblasts, and diverse tissue-resident cell types [28]. We employed the TIMER 2.0 tool to assess immune cell infiltration in breast cancer tissues using RNA-Seq expression profile data from the TCGA database. These data were analyzed alongside patients’ clinical information for further prognostic insights. The results revealed that among the six immune cell types, macrophages exerted the most pronounced impact on patient prognosis (Figure 2A).

Additionally, in the TCGA dataset, a higher level of macrophage infiltration correlated with reduced overall survival time for breast cancer patients (Figure 2B), including those with luminal breast cancer (Figure 2D). Clinical specimens of 59 breast cancer patients collected from the hospital also demonstrated that the median survival time in the high CD68^+^ macrophage infiltration group (1176 days) was lower than that in the low infiltration group (1740 days) (Figure 2C). In order to further study the impact of macrophages on the prognosis of breast cancer patients under different HR conditions, we used the TCGA database for analysis. In patients with hormone-receptor-positive breast cancer, higher macrophage infiltration was associated with a poorer clinical prognosis. However, this relationship was not observed in patients with hormone-receptor-negative breast cancer (Figure 2E). Furthermore, the connection between estrogen receptor (ER) status, progesterone receptor (PR) status, and clinical prognosis exhibited varying results. The extent of macrophage infiltration had a more pronounced impact on ER^+^ breast cancer and PR^-^ breast cancer (Appendix A). In addition, we also used the TCGA database to analyze the degree of macrophage infiltration and the expression level of HR. We noted that higher levels of macrophage infiltration corresponded to reduced ESR1 or PGR expression in breast cancer (Appendix A) and luminal breast cancer (Appendix A).

### 3.3. Macrophages Induce Changes in Luminal Breast Cancer Hormone Receptors and HER2

To analyze the association between luminal breast cancer receptor conversion and macrophages, we initiated by subjecting luminal breast cancer cells to a conditioned culture medium obtained from macrophages. We used different macrophage molecular markers to verify the success of macrophage differentiation and polarization (Appendix A). Following 24 h exposure to the conditioned culture medium, we noted a transition in the BC cells’ morphology, shifting from agglomerated, epithelial forms to sporadic, mesenchymal structures. This phenomenon became more prominent with higher proportions of conditioned culture medium added (Figure 3A). Then, we conducted assessments of mRNA and protein expression levels of hormone receptors (ER and PR), HER2, and EGFR in breast cancer cells using RT-qPCR and Western blot techniques. After treating breast cancer cells with a macrophage-conditioned medium for 24 h, we observed a down-regulation of ER and PR, coupled with an up-regulation of HER2 and EGFR (Figure 3B,C).

### 3.4. Macrophages Induce Tamoxifen Resistance in Breast Cancer

To investigate the role of macrophages in the development of tamoxifen resistance, we administered a 20% conditioned culture medium from macrophages to T47D or BT474 cells, followed by treatment with 4-HT or fulvestrant. Upon treating T47D or BT474 cells with 4-HT, the CCK-8 assay results revealed that the group treated with macrophages exhibited reduced resistance to 4-HT. This was evident through increased cell viability at different time points (Figure 4A), a higher IC50 value (Figure 4B), and decreased apoptosis levels (Figure 4D,E). Upon treatment with 100 nM fulvestrant, the CCK-8 results showed that the macrophage-treated group had higher cell viability (Figure 4C).

### 3.5. Macrophages May Regulate the Expression of Hormone Receptors and HER2 through the MNX1 Transcription Factor

To investigate the mechanism through which macrophages influence receptor changes in breast cancer cells, we initially identified two potential transcription factors from three datasets that could potentially contribute. These datasets included genes differentially expressed in HER2-E breast cancer compared to Luminal breast cancer, gene sets coexpressed with ERBB2, and transcription factors potentially regulating ERBB2. We screened out two potential transcription factors: MNX1 and NKX2-2 (Figure 5A). HER2 is highly expressed in Luminal B and HER2-E breast cancer. As potential transcription factors regulating the expression of ERBB2, MNX1 and NKX2-2 emerged as the highest-expressed in HER2-E breast cancer, followed by Luminal B breast cancer (Figure 5B). Notably, the expressions of MNX1 and NKX2-2 showed positive correlations with ERBB2 and EGFR, while exhibiting negative correlations with ESR1 and PGR (Figure 5C). Subsequently, we observed that macrophages could regulate the expression of MNX1, but they did not significantly affect NKX2-2 (Figure 5D,E). Moreover, MNX1 was found to be associated with poor clinical outcomes, whereas NKX2-2 did not exhibit an impact on clinical outcomes (Figure 5F,G).

## 4. Discussion

The discordance of receptor status between primary tumors and paired metastases in breast cancer is well recognized [16,17,46]. The discordance rates of ER, PR, and HER2 exhibited considerable variation among breast cancer patients [16,20,22]. The conversion of receptor status from positive to negative was statistically more probable than the reverse transition from negative to positive [16]. Moreover, patients who experienced a shift from negative to positive receptor status performed better than those whose receptors switched from positive to negative [47]. The therapeutic decisions depend on the status of ER, PR, and HER2. Loss of receptor status in primary tumors may result in patients receiving ineffective treatment with associated toxicity risks. Simultaneously, a lack of comprehension regarding the metastases of acquired receptor status may lead to erroneous refusal of effective treatment.

Breast cancer is prone to lymph node metastasis, and lymph node metastasis occurs in the early stage of breast cancer, especially for luminal breast cancer [3]. Luminal tumors represent the most common subtype of breast cancer, and patients in advanced stages often confront endocrine therapy resistance, recurrence, and metastasis. Consequently, we delved into the clinical subtype differences between paired primary and lymph node metastatic breast tumors and made the following observations: (1) the molecular subtype of breast cancer is largely maintained during LNM, especially for luminal A tumors; (2) the proportion of luminal B, HER2, and Basal subtypes increased among all PAM50, while luminal A and Normal subtypes decreased in PLN; and (3) down-regulation of hormone receptors and upregulation of HER2 during lymph node metastasis in the popliteal lymph node metastasis model. Inaccurate receptor conversion rates may made due to the limited sample size, especially in HER2-E and TNBC tumors. Due to the lower incidence of HER2-E and TNBC tumors compared to luminal tumors, we included only a limited sample size of around 12 cases in our study, resulting in a relatively high receptor conversion rate. Increasing the sample size would enhance the accuracy of our findings. The popliteal lymph node metastasis model demonstrated that at least one of ER and PR remained positive, while HER2 protein overexpression occurred in four of eight positive popliteal lymph nodes. The low probability of this transition is consistent with clinical phenomena. Increasing the sample size of animal experiments is conducive to exploring this subtype conversion.

The discrepancy in receptor status between primary breast cancer and metastases may arise from various factors, encompassing alterations in disease biology, tumor heterogeneity, the impact of previous treatments on clonal subsets, the sampling error in focal tumors, and the lousy accuracy or low repeatability of receptor tests and gene amplification assays [16,48,49]. Some meta-analysis studies reported the effect of treatment on HR conversion and the effect of trastuzumab treatment on HER2 conversion [50,51,52,53]. However, some studies showed that this correlation cannot be demonstrated [54,55]. According to an article, tamoxifen resistance in MCF7-TamR (a tamoxifen-resistant cell model) was not caused by mutations or altered expression of ER [56]. We cannot confirm the impact of therapy on HR and Her2 conversion and the causal relationship between drug resistance and receptor conversion. Breast cancer is a highly heterogeneous tumor [5,49]. Luminal breast cancer is now defined by the presence of at least 1% ER^+^ or PR^+^ cells [57]. This raises queries about the origin and biology of receptor-negative cells in luminal tumors. A small neoadjuvant study suggests that anti-estrogenic or aromatase inhibitors increase the number of ER^−^ cells in drug-resistant or relapsing diseases [58]. The results of single-cell sequencing also showed that the whole single-cell expression profile was consistent with the mass tumor expression profile and pathological results. However, individual cells showed heterogeneity of ER (ESR1), PR (PGR), and HER2 (ERBB2) gene expression [59]. The heterogeneity of the primary tumor is closely related to drug resistance [5,58]. The origin of the primary tumor needs further exploration. Therefore, we need to continue exploring the subtype conversion mechanism to achieve early intervention and precise treatments.

The role of TME in dynamically regulating tumor progression and its influence on treatment outcomes is widely recognized [27]. In our study, we found that macrophages exerted the most pronounced impact on patient prognosis among the six immune cell types. The correlation analysis between the degree of macrophage infiltration and receptor expression in the TCGA dataset showed that macrophages potentially wielded an essential role in “receptor conversion” and contributed to the heterogeneity of breast cancer. However, whether macrophage-induced receptor status changes manifest in primary tumors or PLN necessitates further validation. In the mice model, luminal A breast cancer cell line T47D (ER^+^ PR^+^ HER2^−^) was planted in the footpad, expression level discrepancies in ER and PR within primary tumors were evident, while HER2 overexpression emerged in one of eight footpad primary tumors. In addition, it is worth contemplating whether the incongruity of clinical subtypes between primary tumors and PLN is due to heterogeneous tumor cell metastasis from primary to PLN or phenotypic alterations during lymph node metastasis.

Many studies have demonstrated that M1 macrophages exert an anti-tumor function, whereas M2 macrophages promote tumors [60]. In our study, we treated breast cancer cells from different macrophage types with a conditioned culture medium. Interestingly, M0, M1, and M2 macrophages can induce receptor conversion and tamoxifen resistance, which may be due to the role of some cytokines secreted by these three types of macrophages, which is worth further study. In addition, we need to separate macrophages from tumor tissues to verify our conclusion further.

Finally, our study revealed that macrophages upregulated the expression of MNX1 transcription factor, and MNX1 might regulate the expression of hormone receptors and HER2. MNX1 is a homeobox gene known as an oncogene in infant AML [61] and prostate cancer [62,63]. MNX1 also promotes the malignant progression of cervical cancer [64], breast cancer [65], and colorectal cancer [66]. MNX1 expression escalates through AKT signaling independently of mTOR [63]. Overexpression and activation of HER2 can lead to the activation of the PI3K/AKT pathway [67,68,69]. In conclusion, crosstalk may exist between MNX1 and HER2. The regulatory association between MNX1 and the expression of breast cancer receptors remains uncharted. However, the intricate mechanism underlying MNX1 and BC receptor expression warrants further investigation.

## 5. Conclusions

Many of the biological changes that underlie the progression of breast cancer metastasis remain shrouded in mystery. In this context, we undertook a comparison of clinical subtypes within paired primary and lymph node metastatic tissues. Our findings hint that while clinical subtypes tend to persist throughout metastatic progression, luminal tumors can transition into HER2-E BC during this process. Macrophages emerge as wielding the most significant impact on the clinical prognosis of breast cancer among various immune cell types. Upon exposure to a macrophage-conditioned medium, the expression of hormone receptors was observed to diminish, while HER2 expression experienced an upregulation. Macrophages contributed to the promotion of tamoxifen resistance in breast cancer cells. Bioinformatics analysis unveiled MNX1 as a potential key transcription factor responsible for regulating hormone receptor and HER2 expression in macrophages. In essence, our study unveils the shifting clinical subtypes that occur during lymph node metastasis, thereby underscoring the critical relevance of targeted macrophage therapy.

Most of the biological changes that occur during the progression of breast cancer metastasis are largely unknown. In this study, we undertook a comparison of clinical subtypes within paired primary and lymph node metastatic tissues. Our findings hint that although clinical subtypes are largely maintained during metastatic progression, luminal tumors can converse to HER2-E BC during metastatic progression. Macrophages emerge as wielding the most significant impact on the clinical prognosis of breast cancer among various immune cell types. The expression of hormone receptors was down-regulated, and HER2 expression was up-regulated after treatment with a macrophage-conditioned medium. Bioinformatics analysis unveiled MNX1 as a potential key transcription factor for regulating hormone receptor and HER2 expression. Overall, our study reveals the changes in clinical subtypes during lymph node metastasis and highlights the importance of targeted macrophage therapy.

## Figures and Tables

**Figure 1 cancers-16-00678-f001:**
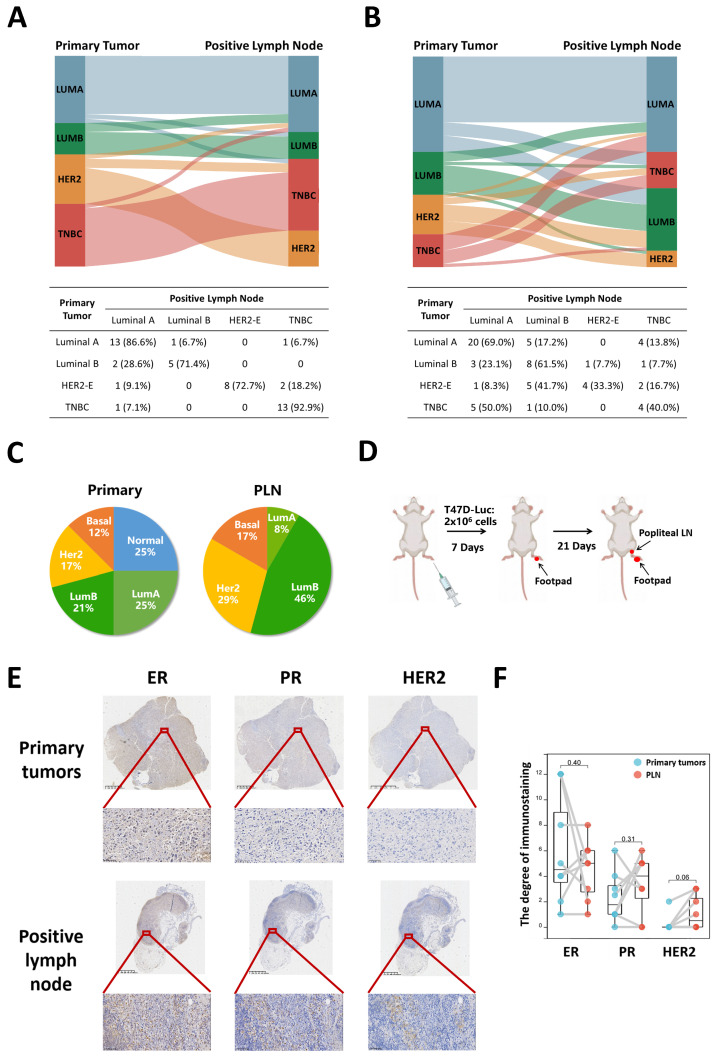
Subtype changes between primary tumors and PLN. (**A**) Subtype concordance between primary tumor and positive lymph node in the tissue microarrays of 47 breast cancer patients. (**B**) Subtype concordance between primary tumor and positive lymph node in 64 breast cancer patients collected from the hospital. (**C**) Distribution of PAM50 intrinsic subtype in primary tumors versus PLN, *n* = 24. (**D**) The schematic picture of the popliteal lymph node metastasis model, *n* = 8. (**E**) The expression of estrogen receptor, progesterone receptor, and HER2 in footpad primary tumor in situ and its paired positive popliteal lymph node of nude mice. Top, 5× magnification (scale bar, 160 μm); bottom, 200× magnification (scale bar, 10 μm), *n* = 8. (**F**) The degree of ER, PR, and HER2 immunostaining in footpad primary tumor in situ and its paired positive popliteal lymph node of nude mice, *n* = 8.

**Figure 2 cancers-16-00678-f002:**
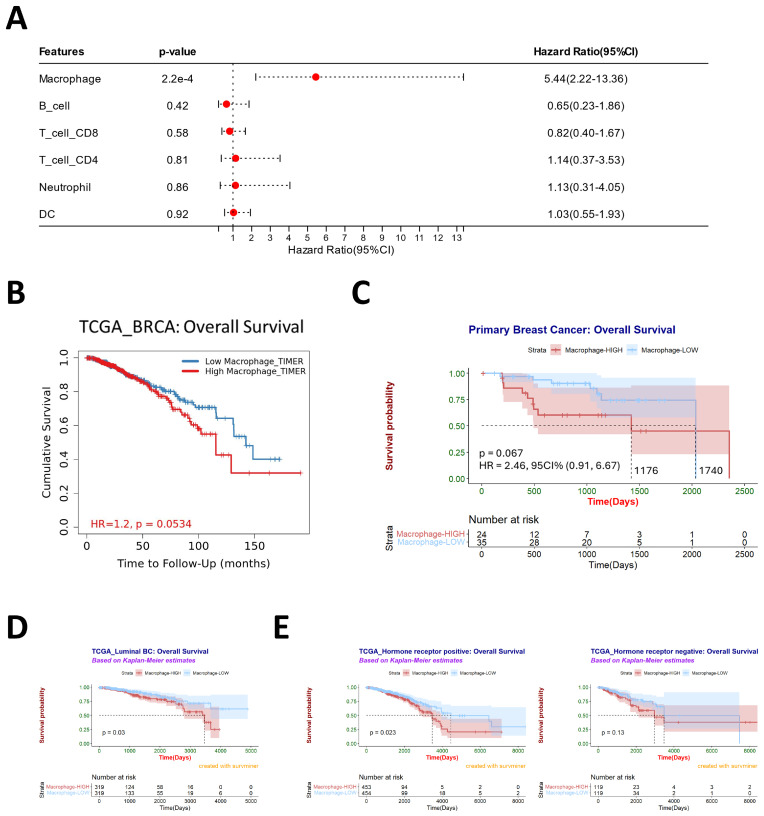
Macrophages play an essential role in breast cancer. (**A**) The multivariate Cox hazards regression analysis of multiple immune cell types in breast cancer. The immune cell infiltration was assessed by TIMER 2.0. Hazard Ratio < 1: reduction in hazard, Hazard Ratio = 1: no effect, and Hazard Ratio > 1: increase in hazard. Data from TCGA. (**B**) Survival analysis of macrophages in breast cancer, data from TCGA, *n* = 1100. TIMER 2.0 performed the Cox Proportional Hazard Model and Kaplan–Meier survival curves. (**C**) The degree of primary CD68^+^ macrophage infiltration and clinical prognosis of 59 breast cancer patients we collected from the hospital were analyzed using the optimal truncation ratio. (**D**) Survival analysis of macrophages in luminal breast cancer, data from TCGA, and the infiltration degree of macrophages were grouped using TIMER 2.0, *n* = 638. (**E**) Survival analysis of macrophages in HR+/HR− breast cancer, data from TCGA, and the infiltration degree of macrophages were grouped using TIMER 2.0, *n* = 907. R 4.3.1 software and KM-Plotter performed Kaplan–Meier survival curves and Log-rank statistical analysis in (**C**,**E**).

**Figure 3 cancers-16-00678-f003:**
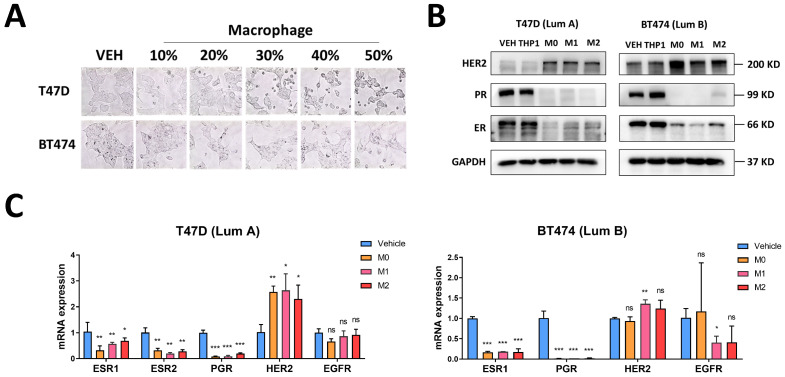
Macrophages induce changes in hormone receptors and HER2. (**A**) Morphological changes of breast cancer cells treated with different gradient M2 macrophages conditioned medium, 200× magnification. (**B**,**C**) Protein and mRNA levels of hormone receptors and HER2 in breast cancer cells treated with macrophage conditioned medium. The original western blot figures can be found in File S1. Data present as mean ± SEM, ns indicates *p* > 0.05, * *p* < 0.05, ** *p* < 0.01, *** *p* < 0.001.

**Figure 4 cancers-16-00678-f004:**
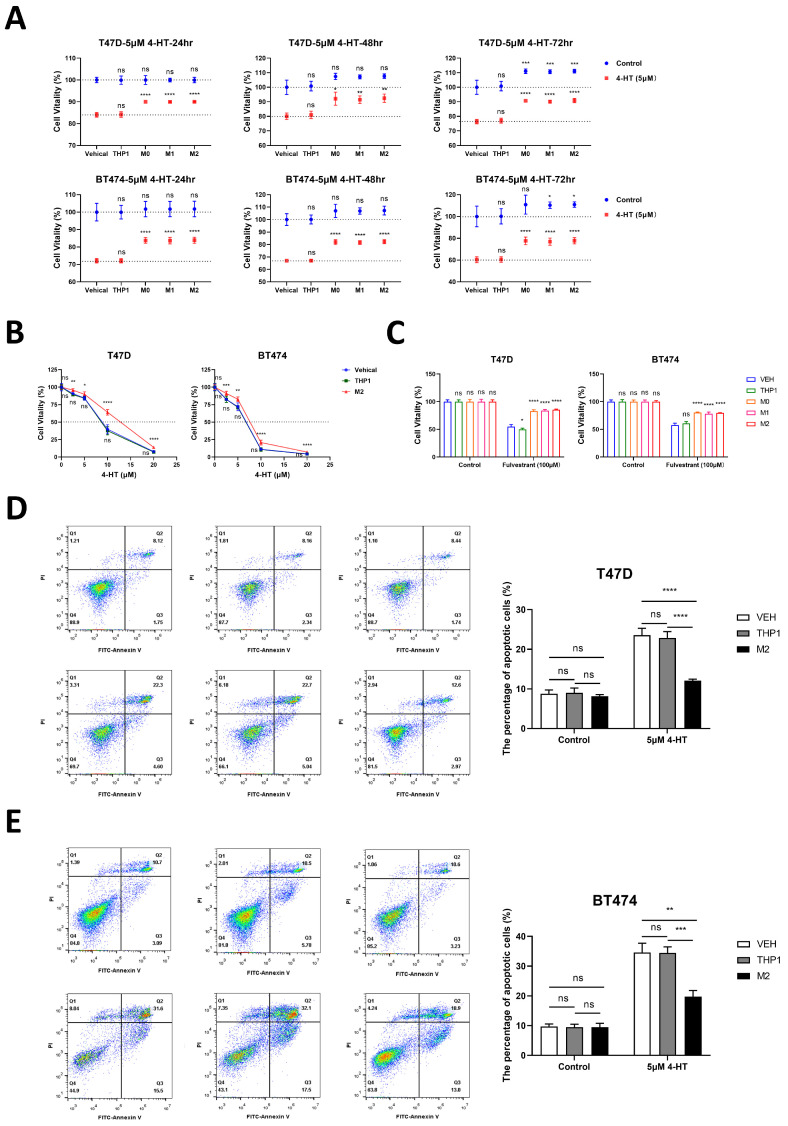
Macrophages induce tamoxifen resistance in breast cancer cells. (**A**) After 5 μM tamoxifen treatment, the cell viability of the group with macrophages or THP1 conditioned medium treatment, and the vehicle group was measured at 24 h, 48 h, and 72 h; each time point was standardized according to the cell viability of the untreated group. (**B**) The cell viability of the vehicle group and the group with macrophages or THP1 conditioned medium treatment was measured under different tamoxifen concentration gradients (0, 2.5, 5, 10, 20 μM) at 24 h. (**C**) After 100 nM fulvestrant treatment, the cell viability of the group with macrophages or THP1 conditioned medium treatment, and the vehicle group was measured at 24 h. (**D**,**E**) After 24 h of treatment with 5 μM tamoxifen, the apoptotic cell rate in the group with macrophages or THP1 conditioned medium treatment and the vehicle group was measured by FCM. Data present as mean ± SEM, ns indicates *p* > 0.05, * *p* < 0.05, ** *p* < 0.01, *** *p* < 0.001, **** *p* < 0.0001.

**Figure 5 cancers-16-00678-f005:**
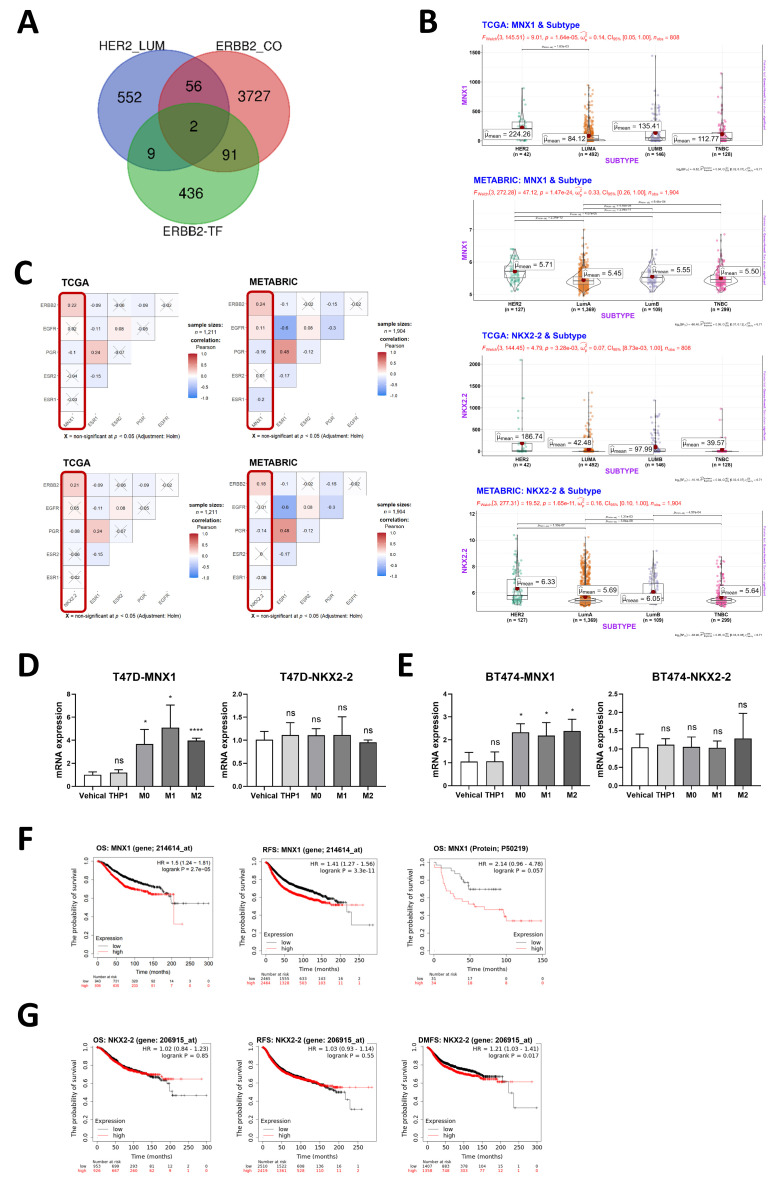
Macrophages may regulate the expression of hormone receptors and HER2 through the MNX1 transcription factor. (**A**) The three data sets from TCGA were differential genes highly expressed in HER2-E breast cancer compared with Luminal breast cancer, gene sets coexpressed with ERBB2, and transcription factors that may regulate ERBB2. (**B**) MNX1 and NKX2-2 gene expression levels in four breast cancer subtypes, data from TCGA and METABRIC. (**C**) Correlation analysis of gene expression levels of 2 transcription factors and ESR1, PGR, ERBB2, and PGR, data from TCGA and METABRIC. (**D**,**E**) mRNA expression levels of MNX1 and NKX2-2 after macrophages or THP1 conditioned medium treatment. Data present as mean ± SEM, ns indicates *p* > 0.05, * *p* < 0.05, **** *p* < 0.0001. (**F**) Survival curve analysis between MNX1 gene and protein expression level and clinical prognosis in breast cancer, data from KM-Plotter. (**G**) Survival curve analysis between NKX2-2 gene expression level and clinical prognosis in breast cancer, data from KM-Plotter.

## Data Availability

All data generated or analyzed during this study are included in this published article and its Appendix A.

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
