# Peer review of "Macrophages Promote Subtype Conversion and Endocrine Resistance in Breast Cancer"

_cancers, 2024, doi:10.3390/cancers16030678_

Round 1

Reviewer 1 Report

Comments and Suggestions for Authors

The study by Zhang et al is overall interesting and well written. 

for publication there is a need to add some more info to the Methods. 1. information on antibodies dilutions used in WB and IHC, 2. how were the samples prepared for IHC and if the primary vs lympho node samples were stained the same way/same day for correct comparison as DAB is not quantitative in principle. 3. needs info on image magnification; 4. define staining index.

Additionally, why did the authors polarize macrophages to M1 with LPS, which mimics a bacterial infection, vs IFNG which would have been a better choice? also, previous reports showed polarization specific effects on ER downregulation but here only differentiation to M0 is sufficient negating a polarization effect which appears important in primary tumors - please comment.

Is the HR and Her2 conversion dependent on the type of treatment? all the patients in the analysis are considered under the same regime which is not stated if it is true and which may alter the interpretation of the clinical data outcome. It is very important to be clear if the patient population analyzed is correctly chosen - for example, what happens in patients that are tamoxifen sensitive vs resistant? or with Herceptin treated patients? the same idea applies to figure 2 - it might be useful to better stratify the patients

Figure 1E hard to see, may need higher mag

Figure 1F shows no stat difference and is not clear what the y axis is

Figure 3 - did the authors treat the cells for longer than 24h and determine what happens to the receptors etc? also, mobility/invasion should be measured

Figure 4 is very confusing - panel A seems to be weirdly normalized - we suggest normalizing to one time point to show that the therapies indeed work in the no CM samples. Also the shift in IC50 is marginal at best and may not be clinically relevant. The assay should be run longer than 72 h and the concentrations of OHT and Fulv are really off - 100uM Fulv is unrealistic in practice and the affinity for the ER is in the pM to nM range for both drugs. Panels D and E should be represented similarly and better labeled, the x axis in panel E contains an error and there are no bars for M0 and M1

the observation about MNX1 is very interesting but it appears preliminary without some follow on experiments (i.e., knock-down)  

Reviewer 2 Report

Comments and Suggestions for Authors

Review: Macrophages promote subtype conversion and endocrine resistance in breast cancer

The authors sought to describe elements of molecular subtype conversion between primary tumors and lymph node metastasis (LNM), which has not yet been explored in depth. They used tumor samples from BC patients as well as human cell lines and mouse models to explore this relationship. They found that macrophages drove the conversion of luminal to HER2-E subtype during LNM and suggest that macrophages may be crucial in targeted treatment for LNM. Overall, the findings are interesting and are important in the implications for treatment. While the authors clearly explored this issue in depth, the results were a bit hard to follow at times. The paper could be improved by providing more clarity on which datasets were being used for which analyses. Additional details need to be provided for adjustment variables in statistical analyses. The discussion should be expanded to speak to each result, and the limitations of the study need to be clearly described.

Individual comments by section are included below:

Introduction

-Line 54: “The reported inconsistencies of ER, PR, and 54 HER2 so far vary widely: 3%-54% for ER, 5%-78% for PR, and 0%-34% for HER2.” The term “inconsistencies” is a bit vague here, I would reword to make it clearer what exactly the percentages being presented are related to (i.e., is this % of tumor cells converting, or the % of tumors that tend to convert?)

-Can the authors provide more background on the evidence for TAM involvement in receptor conversion?

Methods

-Under section 2.5 please include how clinical subtype conversion was defined (i.e., what degree of change in immunostaining was considered a conversion). Was it any change in grading category, or was it considered a change from 0 to 2+, for instance.

-Under section 2.10 please indicate adjustment factors for the survival analysis. Age should at least be considered, as well as initial subtype/treatment.

-Please provide an explanation of the analyses done using the publicly available datasets- SEER/METABRIC; this seems to be missing and makes the first section of the results a bit confusing.

Results

-In general, the figures are well designed; however, it is a bit hard to follow the results in terms of which data source is being used for which analysis, especially in figures where multiple data sources are being used. It would be helpful to more clearly indicate where the data for each figures or subfigure is coming from. If different N’s are given more descriptive terms would be helpful, e.g.: in figure 2C “clinical prognosis of 59 breast cancer patients were analyzed..” – after listing 59 BC patients it would be helpful to describe this patient subset (why this is not the full 64). It may be additionally helpful to separate figures by data source (e.g., BC tumor data in separate figures than cell line data & mouse models) so it is less confusing to follow.

-In the second paragraph it is a bit unclear what data source is being used to explore these distributions- please clarify that this is among the BC patient samples, since the previous paragraph discusses SEER/metabric findings.

-Line 266-268 & lines 295-297, 322-324: These conclusion statements should be part of the discussion as opposed to the results.

-Figure 1: Why is figure 1A out of 47 BC patients and B is out of 64 – it is not clear within the text or within the figure legend what the difference is between these two groups. Fig. 1C is listed in the text before Fig 1A and 1B –the figure or text should be reordered so these appear consecutively.

-Figure 2: For (a) please label what the adjustment variables were for the multivariate analysis.

-Figure 3: Is there a clearer W. blot for T47D samples? The background for ER is a lot, and based on positioning compared to the BT474 it is not clear if the thickest lines are actually representative of ER, or if the thin lines below are closer to 55 KD.

-Figure 5: 5b is very hard to read – can you please increase clarity? Figure 5F and G: Please label y axis as probability of survival

Discussion/Conclusion

-The discussion could be more detailed. It would be good to provide your conclusive statements about each element of the results and then dive into how findings align with biologic plausibility/evidence from other studies. Limitations should be discussed in depth. The conclusion could be tightened to focus on the most significant findings here.

Comments on the Quality of English Language

Quality of English is good, a few wording choices are a bit odd but this will only require minor edits.

Reviewer 3 Report

Comments and Suggestions for Authors

PLease read carefully the instructions fr authors and formatted the references according to these instructions. 

Comments on the Quality of English Language

Minor revisions for English needed

Author Response

Dear reviewer:

Thank you very much for taking the time to review this manuscript. We are sorry for the mistake. We have formatted the references according to the instructions for authors.

Round 2

Reviewer 1 Report

Comments and Suggestions for Authors

thank you for answering all the question I had in a very precise and extensive manner